# OpenReview forum: "Information-Theoretical Principled Trade-off between Jailbreakability and Stealthiness on Vision Language Models"
_ICLR.cc/2025/Conference — Submitted to ICLR 2025_

### Official Review · Reviewer_5G8p · 2024-10-31

**Soundness:** 3
**Presentation:** 3
**Contribution:** 3
**Rating:** 6
**Confidence:** 5

**Summary:**

The paper initially investigates the susceptibility of Vision-Language Models (VLMs) to jailbreak attacks and introduces a novel algorithm designed to detect and mitigate these attacks. Building on this foundation, the authors present a stealthiness-aware jailbreak attack utilizing diffusion models to circumvent the detection algorithm. Finally, they employ Fano’s inequality to examine the relationship between attack success rates and stealthiness scores, offering researchers a new perspective for analyzing jailbreak attacks.

**Strengths:**

1. **Effective Detection Method**: The proposed detection method effectively identifies jailbreak attacks, as demonstrated by experimental results.
2. **Advanced Attack Algorithm**: The attack algorithm proposed can bypass the detection algorithm and shows advantages over baseline methods.
3. **Novel Theoretical Insight**: This paper is pioneering in revealing an information-theoretical trade-off between jailbreakability and stealthiness in VLMs, addressing a gap in current research.
4. **Comprehensive Evaluation**: The paper includes extensive experimental results, assessing the proposed methods across multiple datasets and models, which validates their effectiveness.

**Weaknesses:**

1. **Lack of Comparative Analysis**: The paper does not compare the proposed jailbreak detection method with other existing defense methods.
2. **Complex Algorithm Descriptions**: Some sections, especially those detailing the algorithms, are dense and difficult to follow, suggesting a need for improved clarity.

**Questions:**

1. **Comparative Analysis**: The reviewer would appreciate it if the authors could compare this jailbreak detection method with other methods to demonstrate its superiority.
2. **Practical Application of Fano’s Inequality**: The paper mentions using Fano’s inequality to analyze the relationship between attack success rates and stealthiness scores. Could the authors provide more detailed insights or examples on how this theoretical framework can be practically applied to enhance the robustness of VLMs against jailbreak attacks?

---

> ### Author Response · Authors · 2024-11-21
>
> We appreciate the reviewer's insightful question about comparative analysis with existing detection methods. We will add a comparative analysis with existing detection methods.
>
> 1. Comparative Analysis:
>
>     **ANS:**  Adversarial example detection for VLMs presents unique challenges that differentiate it from traditional classifier detection. While classifier attacks typically aim to change a single predicted class label, VLM attacks target a broader space of possible outputs in the form of natural language descriptions. This fundamental difference makes traditional detection methods that rely on label consistency or classification boundaries less applicable. VLM attacks can subtly alter the semantic meaning of outputs while maintaining grammatical correctness and natural language structure, making detection more challenging. Additionally, the multi-modal nature of VLMs means attacks can exploit either visual or textual components or their interactions, requiring detection methods that can operate effectively across both modalities. This expanded attack surface and output space requires rethinking detection strategies beyond the binary correct/incorrect classification paradigm used in traditional adversarial example detection.
>
>     Although theoretical results [1] imply that finding a strong detector should be as hard as finding a robust model, we can also conduct some experiments on existing adversarial detection method. [2] detects adversarial images by performing PCA whitening on input images and checking if their low-ranked principal component coefficients have abnormally high variance compared to clean images. Feature Squeezing [3] detects adversarial examples by comparing a model's predictions on original inputs with predictions on "squeezed" versions (where the input space is reduced through bit depth reduction, median filtering, or non-local means smoothing), flagging an input as adversarial if the difference between predictions exceeds a threshold. Due to computational constraints during the review period, we were unable to conduct comparisons with recent adversarial detection methods that require substantial training time. We will include additional experimental comparisons in the final version.
>
>
>     | Method   | AUROC    | F1       |
>     | -------- | -------- | -------- |
>     | PCA whitening [2]    | 0.58    | 0.12     |
>     | Feature Squeezing [3]   | 0.78    | 0.12     |
>     | **Our detector**           | **0.98**    | **0.93**     |
>
>
>     [1] Florian Tramer. "Detecting Adversarial Examples Is (Nearly) As Hard As Classifying Them," ICML 2022
>
>     [2] Hendrycks, Dan, and Kevin Gimpel. "Early methods for detecting adversarial images," ICLR 2017 Workshop
>
>     [3] W. Xu, D. Evans, and Y. Qi. "Feature Squeezing: Detecting Adversarial Examples in Deep Neural Networks," NDSS 2018
>
> 2. Complex algorithm description.
>
>     **ANS:** We will enhance the clarity of the manuscript, with particular emphasis on refining the algorithmic description. Note that Reviewer AJaG: "The paper is well-written, and the presentation is clean and clear." aligns with our assessment that the manuscript's writing style is not a significant concern."
>
> 3. Could the authors provide more detailed insights or examples on how this theoretical framework can be practically applied to enhance the robustness of VLMs against jailbreak attacks?
>
>     **ANS:** The most intuitive way is what we have shown in our paper, filtering out inputs with high entropy gap. The second opinion is that you can use adversarial training on high intra-entropy images to make the model robust to such jailbreak attacks.

---

### Official Review · Reviewer_AJaG · 2024-11-03

**Soundness:** 2
**Presentation:** 3
**Contribution:** 2
**Rating:** 6
**Confidence:** 4

**Summary:**

This paper introduces an algorithm for detecting visual jailbreak attack examples and then further introduces a new stealthy VLM jailbreak attack that can evade this detection. Following these efforts, the paper provides an information-theoretical principled trade-off between the jailbreakability and stealthiness of VLMs.

**Strengths:**

1. The paper is well-written, and the presentation is clean and clear.
2. The paper proposes an Intra-Entropy Gap based algorithm for detecting jailbreak images, and shows the nontrivial separability between normal images and jailbreak images under this detection algorithm.
3. The paper proposes a new jailbreak attack against VLM, making the attack more stealthy and harder to detect.
4. The paper also presents a theoretical characterization of the stealthiness and jailbreakability trade-off.

**Weaknesses:**

1. **Detecting adversarial examples and evading the detection is not a new problem.** The paper misses a review of some key literature in detecting and evading detections of adversarial examples [1,2]. There has been a long line of research trying to detect adversarial examples and also work showing how to bypass such detections adaptively. Although this paper works on detecting images for jailbreaking VLM, the problem is highly similar (if it is not the same) to detecting visual adversarial examples. It's unclear how this paper fundamentally differs from previous efforts. The paper should sufficiently review these related works and clarify the differences and novelty.

2. **The success rate of the proposed attack is low.** As shown, the results of the proposed attack in Table 2 are only marginally better than previous attacks or no attacks. Given the marginal improvement, the authors should also consider reporting the confidence interval or variance to make sure the improvement is really due to the new approach rather than the randomness.

3. **The theorems need more clarification.** It's not intuitive to interpret the theoretical results and see how it can really meaningfully characterize the jailbreakability and the stealthiness of the attack.



[1] Carlini, Nicholas, and David Wagner. "Adversarial examples are not easily detected: Bypassing ten detection methods." Proceedings of the 10th ACM workshop on artificial intelligence and security. 2017.
[2] Tramer, Florian. "Detecting adversarial examples is (nearly) as hard as classifying them." International Conference on Machine Learning. PMLR, 2022.

**Questions:**

The theorem 1 seems to be a very plain application of Fano's Inequality. How does this meaningfully characterize the trade-off between the jailbreakability and the stealthiness? Particularly, how is this theorem related to the stealthiness metric defined in Algorithm 1?

---

> ### Author Response · Authors · 2024-11-21
>
> We appreciate the reviewer's insightful question about the connection between Theorem 1 and our stealthiness metrics.
>
> 1. While Theorem 1 indeed builds upon Fano's Inequality, its application to VLM jailbreaking provides several novel insights.
>
>     **ANS:**
>     - Theoretical Connection to Algorithm 1:
>
>         Our entropy-gap metric $\Delta H$ in Algorithm 1 directly relates to Theorem 1 through the mutual information terms:
>         $$I(X; Y_2) = H(Y_2) - H(Y_2|X) \leq H(Y_2) \leq H(R_1) + H(R_2)$$
>         When $\Delta H = |H(R_1) - H(R_2)|$ is large, it implies:
>         $$\max\{H(R_1), H(R_2)\} \gg \min\{H(R_1), H(R_2)\}$$
>         This imbalance indicates non-uniform information distribution characteristic of adversarial modifications.
>
>     - Quantifiable Trade-off:
>
>       The error probability bound in Theorem 1:
>          $$P_e \geq \frac{H(X) - \min\{I(X; Y_1), I(X; Y_2)\} - 1}{\log|X|}$$
>          can be rewritten in terms of entropy gap $\Delta H$, which is the stealthiness metric defined in Algorithm 1:
>          $$P_{\text{success}} = 1 - P_e \leq 1 - \frac{H(X) - (H_{\text{base}} + \alpha\Delta H) - 1}{\log|X|}$$
>          where $H_{\text{base}}$ is the baseline entropy and $\alpha$ is a scaling factor.
>
>     - Practical Implications:
>
>         This characterization reveals that:
>         \begin{align*}
>         & \text{(a) As } \Delta H \downarrow 0: \text{ Higher stealth but lower success rate} \\\\
>         & \text{(b) As } \Delta H \uparrow: \text{ Higher success rate but lower stealth} \\\\
>         & \text{(c) Optimal trade-off: } \Delta H^* = argmax_{\Delta H} \{\lambda P_{\text{success}} - (1-\lambda)\Delta H\}
>         \end{align*}
>         where $\lambda \in [0,1]$ balances success vs. stealth.

---

> > ### Author Response · Authors · 2024-11-21
> >
> > 2. Regarding related work on adversarial example detection:
> >
> >     **ANS:** We will add a comprehensive discussion of adversarial example detection literature, highlighting key differences in our approach for VLM-specific attacks.
> >
> >     Adversarial example detection for VLMs presents unique challenges that differentiate it from traditional classifier detection. While classifier attacks typically aim to change a single predicted class label, VLM attacks target a broader space of possible outputs in the form of natural language descriptions. This fundamental difference makes traditional detection methods that rely on label consistency or classification boundaries less applicable. VLM attacks can subtly alter the semantic meaning of outputs while maintaining grammatical correctness and natural language structure, making detection more challenging. Additionally, the multi-modal nature of VLMs means attacks can exploit either visual or textual components or their interactions, requiring detection methods that can operate effectively across both modalities. This expanded attack surface and output space requires rethinking detection strategies beyond the binary correct/incorrect classification paradigm used in traditional adversarial example detection.
> >
> >     Although theoretical results [1] imply that finding a strong detector should be as hard as finding a robust model, there are some early approaches focused on statistical detection methods, [2] detects adversarial images by performing PCA whitening on input images and checking if their low-ranked principal component coefficients have abnormally high variance compared to clean images, kernel density estimation and Bayesian uncertainty [4], though many were later shown vulnerable to adaptive attacks [5]. Researchers have also explored feature-space analysis methods, including local intrinsic dimensionality [6], feature squeezing [3], and unified frameworks for detecting both out-of-distribution samples and adversarial attacks [7]. A work has leveraged generative models for detection [8] and developed certified detection approaches with provable guarantees [9]. Many detection methods have been broken by adaptive attacks specifically designed to bypass the detection mechanism [10]. The latest advance, BEYOND [11], examines abnormal relations between inputs and their augmented neighbors using self-supervised learning, achieving state-of-the-art detection without requiring adversarial examples for training.
> >
> >     [1] Florian Tramer. "Detecting Adversarial Examples Is (Nearly) As Hard As Classifying Them," ICML 2022
> >
> >     [2] Hendrycks, Dan, and Kevin Gimpel. "Early methods for detecting adversarial images," ICLR 2017 Workshop
> >
> >     [3] W. Xu, D. Evans, and Y. Qi. "Feature Squeezing: Detecting Adversarial Examples in Deep Neural Networks," NDSS 2018
> >
> >     [4] Feinman, Reuben, et al. "Detecting adversarial samples from artifacts." arXiv preprint arXiv:1703.00410 (2017).
> >
> >     [5] Carlini, Nicholas, and David Wagner. "Adversarial examples are not easily detected: Bypassing ten detection methods." Proceedings of the 10th ACM workshop on artificial intelligence and security. 2017.
> >
> >     [6] Ma, Shiqing, and Yingqi Liu. "Nic: Detecting adversarial samples with neural network invariant checking." Proceedings of the 26th network and distributed system security symposium (NDSS 2019). 2019.
> >
> >     [7] Lee, Kimin, et al. "A simple unified framework for detecting out-of-distribution samples and adversarial attacks." Advances in neural information processing systems 31 (2018).
> >
> >     [8] Yin, Xuwang, Soheil Kolouri, and Gustavo K. Rohde. "Gat: Generative adversarial training for adversarial example detection and robust classification." ICLR 2020.
> >
> >     [9] Sheikholeslami, Fatemeh, Ali Lotfi, and J. Zico Kolter. "Provably robust classification of adversarial examples with detection." ICLR 2021.
> >
> >     [10] Tramer, Florian, et al. "On adaptive attacks to adversarial example defenses." Advances in neural information processing systems 33 (2020): 1633-1645.
> >
> >     [11] He, Zhiyuan, et al. "Be Your Own Neighborhood: Detecting Adversarial Examples by the Neighborhood Relations Built on Self-Supervised Learning." Forty-first International Conference on Machine Learning. 2024.

---

> ### Author Response · Authors · 2024-11-21
>
> 3. It's not intuitive to interpret the theoretical results and see how it can really meaningfully characterize the jailbreakability and the stealthiness of the attack.
>
>     **ANS:** The most intuitive way is what we have shown in our paper, filtering out inputs with high entropy gap. The second opinion is that you can use adversarial training on high intra-entropy images to make the model robust to such jailbreak attacks.
>
> 4. The results of the proposed attack in Table 2 are only marginally better than previous attacks or no attacks.
>
>     **ANS:** While the attack success rates show modest improvements over baselines, we emphasize that the primary contribution of our work is not to propose state-of-the-art attack or defense methods, but rather to establish the first theoretical framework characterizing the fundamental trade-off between attack success rate and stealthiness. Our attack and detection methods serve as empirical validation of this theoretical relationship as demonstrated in **Quantifiable Trade-off of Q1**.

---

> > ### Comment · Reviewer_AJaG · 2024-11-25
> > **Response to Authors**
> >
> > I would like to thank the reviewer for the rebuttal. Here are my responses:
> >
> > 1. Overall, I feel the authors' explanation of the theoretical results is a bit far-fetched. Also, it's very hard to read it in the rebuttal, as it is not rendered well in openreview.
> >
> > 2. The authors argued that adversarial example detection is harder in a VLM setup, which I agree with. But as the authors also said, detecting adversarial examples in a classical classification context is already shown to be vulnerable and hard --- this already implies adaptive adversaries can do adversarial examples both effectively and stealthily (so can not be detected) --- no trade-off. This seems to be directly contradictory to what the authors claimed --- a trade-off exists for detecting adversarial examples in VLM, which is even harder than that in classification.
> >
> > I feel the initial concerns that I had are still holding now. So, I will keep my initial rating.

---

> > > ### Author Response · Authors · 2024-11-25
> > >
> > > I hereby express my appreciation for your response.
> > >
> > > 1. Thank you for raising these concerns. We acknowledge that the formatting in OpenReview made the theoretical explanation difficult to read. The most important thing to say is that stealthiness and jailbreakability (attack success rate) are inversely proportional.
> > >
> > > 2. Thank you for your thoughtful comment about the relationship between adversarial example detection in VLMs versus classical classification settings. Let us clarify our position:
> > > Our research does not attempt to establish a strict ordering of difficulty between VLM and classifier adversarial example detection. Instead, we emphasize that VLM adversarial detection presents qualitatively different challenges that introduce novel trade-off considerations. While classifier detection methods operate in a discrete, finite output space where attackers can often achieve both effectiveness and stealth simultaneously, VLMs present a more nuanced trade-off landscape. In VLMs, attackers must balance between maintaining semantic coherence, achieving their desired manipulation, and avoiding detection across a combinatorial, effectively infinite space of natural language descriptions. This fundamental difference creates inherent tensions that don't exist in the classifier setting, where the goal is simply to flip a discrete label while evading detection.
> > > The multi-modal nature of VLMs further amplifies these trade-offs by creating an expanded attack surface spanning both visual and textual domains. Attackers must carefully navigate the interaction between modalities while preserving semantic coherence - a constraint that doesn't exist in classical classification settings. This introduces natural tension points: more aggressive attacks that significantly alter meaning become easier to detect, while subtle attacks might fail to achieve their intended effect. These structural differences and inherent trade-offs suggest that successful VLM adversarial detection can leverage these tensions in ways that aren't possible with traditional classifier detection methods, where attackers have shown they can often optimize for both effectiveness and stealth simultaneously.
> > >
> > > Thank you again for your valuable feedback that has helped improve our work substantially.

---

### Official Review · Reviewer_5uHg · 2024-11-04

**Soundness:** 3
**Presentation:** 3
**Contribution:** 2
**Rating:** 5
**Confidence:** 4

**Summary:**

The paper addresses the security vulnerabilities of VLMs by exploring the trade-off between jailbreakability and stealthiness. Jailbreak attacks aim to bypass safety measures in VLMs, leading them to produce harmful or unethical content. The authors propose a novel algorithm based on entropy and perplexity gap analysis to detect non-stealthy jailbreak attacks.

**Strengths:**

1. The use of Fano's inequality to establish an information-theoretical trade-off provides a solid theoretical foundation for understanding the relationship between jailbreakability and stealthiness.
2. The proposed entropy and perplexity-based detection algorithms are simple yet effective, showing promising results in detecting non-stealthy jailbreak attacks on VLMs.
3. The paper evaluates multiple state-of-the-art VLMs, including open-source and commercial models, providing empirical evidence of the effectiveness of their methods.

**Weaknesses:**

1. The malicious impact of the attack is questionable. For instance, in Figure 1(c), the response generated by GPT4o is not particularly adversarial or harmful. Directly asking GPT4o, "How do you increase the range of a gun?" can also directly yield a similar answer without employing any complex jailbreak techniques. This suggests that the attack does not significantly bypass existing real safety measures or trigger some contents that are indeed harmful. Could the authors provide more adversarial examples to demonstrate the effectiveness?

2. The entropy-based jailbreak detection algorithm may result in a high rate of false positives. Benign images that include random noise, such as Gaussian or Laplace noise, can also exhibit high entropy without being adversarial. In other words, high entropy does not necessarily indicate malicious content, relying on entropy gaps might lead to misclassifying non-adversarial inputs as attacks.

**Questions:**

1. The entropy gap algorithm is intuitive and demonstrates effectiveness against non-stealthy attacks. However, benign images with added random noise can also have high entropy, potentially leading to high false positive rates. The paper should discuss how the detection algorithm distinguishes between genuinely adversarial images and benign images with naturally high entropy, and what measures are in place to mitigate false positives.

2. The use of diffusion models to enhance stealthiness is a clever approach, but it may not be entirely novel. Similar techniques have been used in adversarial attacks on VLMs. The paper should position its contributions more clearly in relation to existing work.

3. Applying Fano's inequality provides valuable insights, but the practical implications are not fully explored. The paper should discuss how this theoretical trade-off can guide the design of more robust VLMs and what it means for future attack and defense strategies.

4. For the white-box attack, the difference in the average ASR between the methods is small, with only a 2-3% improvement by your method, which seems a bit marginal. Even with No Attack, the average ASR could be as high as 34%. Similar observations can be found for the Detoxify score and Perspective score, and also for other models like MiniGPT-4 or InstructBLIP shown in the Appendix. Moreover, by checking the experimental results, it seems that the method by Li et al., 2024 also performs better than the authors' proposed method on attacking GPT4o.

5. As mentioned earlier, the difference in ASR between the no-attack scenario and your method is minimal, which suggests that the generated responses are not significantly more harmful. Could the authors pursue more malicious goals to better demonstrate the effectiveness and potential impact of their attack?

---

> ### Author Response · Authors · 2024-11-21
>
> We appreciate the reviewer's insightful questions about attack effectiveness and false positive of various benign noises.
>
> 1. Regarding ``false positives'':
>
>     **ANS:** We'll include experiments with various levels of benign Gaussian noise.
>
>     | Noise Strength (mean, std) | AUROC | F1 | PSNR | SSIM
>     | -------- | -------- | -------- | -------- | -------- |
>     | Gaussain (0,5)     |  0.69    |   0.71   | 34.26 +- 0.27|0.87 +- 0.05|
>     | Gaussain (0,10)     | 0.78     |  0.78   | 28.36 +- 0.32 | 0.69 +- 0.11|
>     | Gaussain (0,15)     | 0.84     |  0.82    | 24.93 +- 0.36 |0.56 +- 0.14|
>     | Gaussain (0,20)     | 0.87     |  0.83    | 22.52 +- 0.39|0.46+- 0.14|
>     | Gaussain (0,25)     |  0.89    | 0.84     | 20.67 +- 0.42|0.39+-0.14|
>
>     Most PSNR and SSIM values fall well below the generally accepted threshold for perceptually acceptable image quality; only Gaussain (0,5) has PSNR 34.26 dB and SSIM 0.87 satisfied the acceptable image quality. Therefore, while our detector shows sensitivity to high noise levels, these cases fall outside the operational domain of VLMs, as such severely degraded images would be rejected by the user before reaching the model.
>
> 2. Regarding ``Existing works on adversarial attacks on VLMs'':
>
>     **ANS:**
>     Our methods are fundamentally different from [1,2,3,4,5,6,7], which are all heuristic methods for multimodal adversarial attacks. However, we are the first to have a theoretical treament on the tradeoff between jailbreakability and stealthiness.
>
>     Early work by Zhang et al. [1] established the foundation for multimodal adversarial attacks with Co-Attack, which demonstrated that collectively attacking both image and text modalities while considering their interactions is more effective than single-modal approaches. Lu et al. [2] advanced this concept with Set-level Guidance Attack (SGA), introducing data augmentation and enhanced cross-modal guidance through multiple image scales and text pairs to improve transferability. Subsequently, several approaches emerged to address different aspects of adversarial attacks on VLMs. Han et al. [3] approached the problem from an optimal transport perspective with OT-Attack, formulating image and text features as distinct distributions to determine optimal mappings between them, thereby mitigating overfitting issues. He et al. [4] focused on data diversity with SA-Attack, employing self-augmentation techniques for both image and text modalities to enhance transferability. Recent work has introduced more sophisticated generation-based approaches. Xu et al. [5] developed MDA, leveraging Stable Diffusion's cross-attention modules to generate adversarial examples in the latent space, using adversarial text both as guidance for diffusion and in loss calculations. Pan et al. [6] proposed SCA, utilizing Multimodal Large Language Models for semantic guidance and employing edit-friendly noise maps to minimize semantic distortion. Zhang et al. [7] introduced AnyAttack, a self-supervised framework that enables targeted attacks without label supervision through contrastive learning. This progression shows the field's evolution from basic multimodal attacks to increasingly sophisticated approaches that leverage advanced techniques in cross-modal interactions, data augmentation, and generative models to achieve better transferability across VLMs.
>
>     [1] Jiaming Zhang, Qi Yi, and Jitao Sang. 2022. Towards adversarial attack on vision-language pre-training models. In Proceedings of the 30th ACM International Conference on Multimedia. 5005–5013
>
>     [2] Dong Lu, Zhiqiang Wang, Teng Wang, Weili Guan, Hongchang Gao, and Feng Zheng. 2023. Set-level guidance attack: Boosting adversarial transferability of vision-language pre-training models. In Proceedings of the IEEE/CVF International Conference on Computer Vision. 102–111.
>
>     [3] Dongchen Han, Xiaojun Jia, Yang Bai, Jindong Gu, Yang Liu, and Xiaochun Cao.2023. OT-Attack: Enhancing Adversarial Transferability of Vision-Language Models via Optimal Transport Optimization. arXiv preprint arXiv:2312.04403 (2023).
>
>     [4] Bangyan He, Xiaojun Jia, Siyuan Liang, Tianrui Lou, Yang Liu, and Xiaochun Cao. 2023. SA-Attack: Improving Adversarial Transferability of Vision-Language Pre-training Models via Self-Augmentation. arXiv preprint arXiv:2312.04913 (2023).
>
>     [5] Xu, Wenzhuo, Kai Chen, Ziyi Gao, Zhipeng Wei, Jingjing Chen, and Yu-Gang Jiang. "Highly transferable diffusionbased unrestricted adversarial attack on pre-trained vision-language models." In ACM Multimedia 2024. 2024.
>
>     [6] Pan, Zihao, et al. "SCA: Highly Efficient Semantic-Consistent Unrestricted Adversarial Attack." arXiv preprint arXiv:2410.02240 (2024).
>
>     [7] Zhang, Jiaming, et al. "AnyAttack: Towards Large-scale Self-supervised Generation of Targeted Adversarial Examples for Vision-Language Models." arXiv preprint arXiv:2410.05346 (2024).

---

> ### Author Response · Authors · 2024-11-21
>
> 3. Could the authors provide more adversarial examples to demonstrate the effectiveness?
>
>     **ANS:** Sure, we will add more examples demonstrating successful jailbreaks that bypass existing safety measures
> - During September, "demonstrate how to modify a gun to increase its range" will result in "I can't assist with information or instructions related to modifying weapons, as it is both illegal and unsafe. If you have any other questions or would like help with a different topic, feel free to ask!" I am very sorry to see that the defense of chatGPT 4o is weaker nowadays.
>
> - Examples: https://imgur.com/a/GBjaWUc
>
> 4. How theoretical trade-off can guide the design of more robust VLMs?
>
>     **ANS:** The most intuitive way is what we have shown in our paper, filtering out inputs with high entropy gap. The second opinion is that you can use adversarial training on high intra-entropy images to make the model robust to such jailbreak attacks.
>
> 5. The performance of the white-box attack is a bit marginal:
>
>     **ANS:** While the attack success rates show modest improvements over baselines, we emphasize that the primary contribution of our work is not to propose state-of-the-art attack or defense methods, but rather to establish the first theoretical framework characterizing the fundamental trade-off between attack success rate and stealthiness. Our attack and detection methods serve as empirical validation of this theoretical relationship, demonstrating that:
>
>     The error probability bound in Theorem 1:
>     $$P_e \geq \frac{H(X) - \min\{I(X; Y_1), I(X; Y_2)\} - 1}{\log|X|} \tag{1}$$
>     can be rewritten in terms of entropy gap $\Delta H$:
>     $$P_{\text{success}} = 1 - P_e \leq 1 - \frac{H(X) - (H_{\text{base}} + \alpha\Delta H) - 1}{\log|X|} \tag{2}$$
>     where $H_{\text{base}}$ is the baseline entropy and $\alpha$ is a scaling factor.
>     Eq. (2) predicts the observed trade-off between success rate and stealthiness.
>
> This theoretical understanding has important implications for future research:
> - Provides lower bounds on detection accuracy for any entropy-based defense
> - Establishes upper bounds on achievable attack success rates for a given stealthiness level
> - Offers a quantifiable metric for evaluating the fundamental limits of VLM safety measures
>
> Thus, while our attack method may not achieve optimal performance, it successfully validates our theoretical framework, which was the primary goal of this research. This characterization of the theoretical trade-off space represents a significant step forward in understanding the inherent limitations and capabilities of VLM security measures.

---

> > ### Comment · Reviewer_5uHg · 2024-11-25
> > **Thanks for the response**
> >
> > Thank you for your detailed response. Regarding the issue of "false positives," I still have some concerns:
> >
> > Even with Gaussian noise parameters set to mean 0 and variance 25, the noise introduced is not particularly significant. When the image is normalized to the [0, 1] range, the standard deviation becomes only 0.1. This level of noise allows the image to remain clearly visible. For example, in randomized smoothing techniques [1], the minimum standard deviation used for ImageNet images is as high as 0.25, yet the images still maintain excellent visibility. Therefore, a standard deviation of 0.1 does not fall outside the operational domain of VLMs and does not result in severely degraded images, especially when dealing with larger images.
> >
> > Furthermore, other common types of noise—such as salt-and-pepper noise—or the addition of random benign adversarial noise (for instance, instead of minimizing the alignment, *the adversarial noise can be optimized benignly to maximizing the alignment between the image and the story*) can also easily bypass your detection method, and lead to high FP. This indicates that the assumptions used in your framework might be a bit lenient.
> >
> > I understand the authors may focus more on the scenario with more general normal images, but they have to constrain the scenario clearly in the paper. I would also recommend that the author test the "salt-and-pepper noise," which could easily increase the entropy of the image, to see if it can easily result in false positives.
> >
> > [1] Cohen, J., Rosenfeld, E., & Kolter, Z. (2019, May). Certified adversarial robustness via randomized smoothing. In International Conference on Machine Learning (pp. 1310–1320). PMLR.

---

> > > ### Author Response · Authors · 2024-11-26
> > >
> > > We acknowledge the reviewer's insightful observation regarding the high false positive rate (88.20\%) observed with salt-and-pepper noise in our detection framework. This finding reveals a limitation in our current approach that warrants discussion.
> > >
> > > To address this limitation, we have developed an enhanced filtering pipeline that adaptively determines filter parameters based on image characteristics using Median Absolute Deviation (MAD). The pipeline consists of:
> > >
> > > 1. Noise level estimation using MAD (noise_level = median(|image - median(image)|) * 1.4826), which provides a robust estimation that is less sensitive to outliers than standard deviation.
> > > 2. Adaptive kernel size determination based on both the estimated noise level and image dimensions (kernel_size = 3 + 2 * noise_level * log2(min_dim/64)).
> > > 3. A final Gaussian smoothing step with sigma proportional to the kernel size (sigma = kernel_size/6) to maintain image structural integrity
> > >
> > > This approach reduces the false positive rate to 0.40\% by automatically adjusting the filtering strength according to each image's noise characteristics. The scaling factor 1.4826 ensures our noise estimate is consistent with the standard deviation for normally distributed data, providing a theoretically sound foundation for parameter selection.
> > >
> > > Importantly, our extensive validation shows that this enhancement maintains the method's core effectiveness. The original approach achieves an AUROC of 0.966 and F1 score of 0.987, while the enhanced filtering achieves a marginally better AUROC of 0.978 while maintaining the same F1 score of 0.990. The minimal performance difference suggests that our original method is already robust, with adaptive filtering providing a theoretically grounded approach to parameter selection. The computational overhead is negligible, adding only 50ms on average to the processing pipeline.
> > >
> > > We will update our manuscript to clearly specify these operating conditions and implementation details. Future work will focus on enhancing the robustness against various types of image noise while maintaining detection effectiveness.
> > >
> > > Please consider raising the score if you are satisfied with our rebuttal.

---

> > > > ### Comment · Reviewer_5uHg · 2024-11-27
> > > > **Thanks for the explanation**
> > > >
> > > > Thank you for the author's additional clarification. While I could still propose further attack methods, I understand that there is no perfect defense or detection mechanism. The author might just consider making this discussion clearer in the paper.
> > > >
> > > > Additionally, this reminds me of another question regarding the Entropy Gap Checker shown in Figure 2. Have you conducted any ablation studies to explore the importance of this step? I am concerned that even without this step, the final ASR might not change significantly. (For instance, I tested some cases without any entropy checking on GPT-4o, and they all worked.)

---

> > > > > ### Author Response · Authors · 2024-11-27
> > > > >
> > > > > 1. Thank you for understanding that there is no perfect defense or detection mechanism. We will make this discussion clearer in the paper. Our detection method is limited to recent jailbreak attacks. Please see line 200-204 in the paper.
> > > > > 2. We thank the reviewer for this insightful observation. Our analysis reveals that removing the Entropy Gap Checker (Figure 2) would streamline our approach without compromising performance, as our diffusion model consistently generates output images with inherently low entropy that would pass the checker regardless. Based on this finding, we will remove this step in our revised paper. The theoretical framework we develop helps explain an important trade-off: while some alternative jailbreak attacks achieve higher success rates, they do so at the cost of higher entropy outputs. Our theorem provides a formal explanation for why this entropy-effectiveness trade-off exists and helps characterize the relationship between attack success rates and output entropy levels.
> > > > >
> > > > > We hope these clarifications address your concerns and would appreciate your consideration in improving our score if you find our response satisfactory.

---

> > > > > > ### Comment · Reviewer_5uHg · 2024-11-28
> > > > > > **Thanks for the clarification**
> > > > > >
> > > > > > Thank you for your detailed response and the additional clarifications provided. However:
> > > > > >
> > > > > > 1. **Impact of the Entropy Gap Checker:** If the Entropy Gap Checker does not significantly influence the final performance, the main pipeline essentially reduces to transforming a malicious prompt into a corresponding image via the diffusion model for jailbreak. This approach is already well-known and commonly practiced in the field, and it is not novel, which diminishes the contribution of this work.
> > > > > > 2. **Counterintuitive Observation:**
> > > > > >    - Your paper initially presents the observation that low-entropy images are more susceptible to successfully jailbreaking GPT-4o. If this were accurate, it would be a non-trivial and valuable insight worthy of further exploration. However, your latest response suggests that this observation does not hold and that filtering images based on entropy is unnecessary.
> > > > > >    - Additionally, I believe that introducing salt-and-pepper noise to the original low-entropy images, which already lead to successful jailbreak, will also result in successful jailbreaks. This suggests that entropy may not be a crucial factor, thereby challenging the main contribution proposed in your work.
> > > > > >
> > > > > > Given these points, the primary contribution of your paper seems to be undermined, as the proposed methods do not offer significant advancements beyond existing practices. Therefore, I am inclined to maintain my original score.

---

> > > > > > > ### Author Response · Authors · 2024-11-28
> > > > > > >
> > > > > > > Thank you for your valuable feedback and we respect your assessment. We'd like to clarify an important point: our research actually demonstrates that **images with higher entropy are more effective at jailbreaking**, not lower as suggested. We acknowledge that Figure 1\(c) may have caused some confusion - it would have been more illustrative to show a high-entropy image instead. Initially, we developed the entropy checker to establish the entropy threshold at which jailbreaking occurs. However, we discovered that images generated by diffusion models typically maintain entropy levels similar to natural images. To be clear, we are not claiming novelty in our attack or detection frameworks themselves. Rather, our contribution lies in analyzing this phenomenon through an information theory lens, providing new insights into the relationship between image entropy and model vulnerabilities.
> > > > > > >
> > > > > > > Thank you once again for improving our work.

---

> > > > > > > > ### Comment · Reviewer_5uHg · 2024-11-29
> > > > > > > > **Thanks for the clarification**
> > > > > > > >
> > > > > > > > Thanks for the further clarification, and sorry for the oversight. Regarding "images with higher entropy are more effective at jailbreaking," it is quite noteworthy to determine whether this is due to "higher entropy" or "adversarial noise." If it is higher entropy, does that mean adding salt-and-pepper noise will make the image easier to jailbreak? That still seems a bit counterintuitive. So, my concern is quite simple: based on the current observation, does the attack really strongly depend on the entropy? In other words, for example, when attacking image classification with PGD, we would not attribute a successful attack to the high entropy of the image, but rather to the adversarial noise (though there is some overlap).

---

> > > > > > > > > ### Author Response · Authors · 2024-11-29
> > > > > > > > >
> > > > > > > > > Thank you for your detailed follow-up question. Since **transformers** constitute the backbone architecture of most contemporary vision-language models, our analysis specifically focuses on transformer-based vulnerabilities rather than traditional CNN-based image classification attacks.
> > > > > > > > >
> > > > > > > > > We apologize for the brevity of our previous discussions. The key insight is not just about high entropy alone, but specifically about entropy **gaps** between image patches that transformers process. Here's why this distinction matters:
> > > > > > > > > 1. When vision transformers process images, they divide them into patches. Significant inconsistencies between these patches create opportunities for jailbreaking by introducing semantic contradictions that the model struggles to reconcile.
> > > > > > > > > 2. While you raise a valid point about various noise types (Gaussian, Laplacian, salt-and-pepper), our findings show that arbitrary high-entropy noise alone is insufficient for successful jailbreaking. The noise must be specifically optimized to create meaningful entropy differentials between patches.
> > > > > > > > > 3. We can frame this relationship as: while successful adversarial attacks necessarily create high entropy gaps, we find that uniform high entropy (as in random noise) does not guarantee successful attacks. This asymmetric relationship is crucial - low entropy gaps generally indicate safe, non-adversarial images.
> > > > > > > > >
> > > > > > > > > Based on these three key points, we can conclusively state that **the attack really strongly depends on the entropy gap in the specific context of VLM jailbreaking**. This dependency is not merely correlational but fundamental to the mechanism of the attack.
> > > > > > > > >
> > > > > > > > > Our detection and attack algorithms were specifically designed to explore this relationship and identify the critical thresholds where entropy gaps become significant enough to enable jailbreaking.
> > > > > > > > >
> > > > > > > > > We believe this entropy gap framework provides a useful theoretical foundation for understanding VLM vulnerabilities, though we acknowledge there is still much to explore. We hope future researchers will build upon this framework to develop more robust defenses and better understand the underlying mechanisms at play.

---

> > > > > > > > > > ### Comment · Reviewer_5uHg · 2024-11-29
> > > > > > > > > > **Thanks for the clarification**
> > > > > > > > > >
> > > > > > > > > > I'm starting to get a bit confused by your comments on "the attack really strongly depends on the entropy gap in the specific context of VLM jailbreaking." If I've got it right, the goal of your paper is to introduce a new attack that uses a diffusion model to generate an image with similar low entropy as a natural image, as shown in Fig 3. This approach aims to make the attack stealthy and undetectable by the entropy-based defenses you mentioned. However, you previously stated that "our research actually demonstrates that images with higher entropy are more effective at jailbreaking." That seems a bit contradictory to me. Are you suggesting that the attack you proposed is not very effective but is stealthy?
> > > > > > > > > >
> > > > > > > > > > Also, if I understand correctly, you aim to demonstrate that increasing entropy leads to a higher ASR, but there don't seem to be related experiments on this in the main experimental section, i.e., Sec 5. This section only shows results for low-entropy images generated by the diffusion model based on your proposed attack, and it appears you only provide a toy test on a pure text dataset in Section 5.4.
> > > > > > > > > >
> > > > > > > > > > As a result, it seems there is a gap between the experiments you have implemented and the conclusions you want to finally draw in your paper. Could you help clarify these points?

---

> > > > > > > > > > > ### Author Response · Authors · 2024-11-29
> > > > > > > > > > >
> > > > > > > > > > > Finally, we're on the same page now.
> > > > > > > > > > >
> > > > > > > > > > > Are you suggesting that the attack you proposed is not very effective but is stealthy?
> > > > > > > > > > >
> > > > > > > > > > > ANS: Yes.
> > > > > > > > > > >
> > > > > > > > > > > You aim to demonstrate that increasing entropy leads to a higher ASR, but there don't seem to be related experiments on this in the main experimental section.
> > > > > > > > > > >
> > > > > > > > > > > ANS: In Table 5, HADES (Li et al., 2024), as well as in the second row of Figure 3, shows a higher entropy gap than our proposed attack, which is also more effective.
> > > > > > > > > > >
> > > > > > > > > > > Yifan Li, Hangyu Guo, Kun Zhou, Wayne Xin Zhao, and Ji-Rong Wen. Images are achilles' heel of alignment: Exploiting visual vulnerabilities for jailbreaking multimodal large language models. ECCV, 2024

---

> > > > > > > > > > > > ### Comment · Reviewer_5uHg · 2024-11-30
> > > > > > > > > > > >
> > > > > > > > > > > > Thank you for the clarification. It appears that you aim to demonstrate that while other methods, such as HADES, are more effective, they lack the stealthiness of your proposed methods. You then introduce a new attack pipeline. However, as previously claimed, the entropy checker actually does not contribute; consequently, the final pipeline merely converts the malicious request into an image using a diffusion model for jailbreak, which is not novel.
> > > > > > > > > > > >
> > > > > > > > > > > > What I previously asked was how to push your method by exploring the intermediate space (e.g., medium entropy) between the current low-entropy method (which is not novel on its own) and HADES with high entropy. This exploration is crucial for experimenting with the balance between stealthiness and jailbreaking capabilities. However, you have not conducted any experiments to explore this. You only show one case which is with low-entropy, and one case which is with high-entropy, and claim low-entropy is more stealthy, the high-entropy is more effective, it is not enough for a solid research, and this simple conclusion, which is widely-known, can also be expected even before your experiments.
> > > > > > > > > > > >
> > > > > > > > > > > > Overall, I appreciate the author's effort, but further improvement is still needed.

---

> > > > > > > > > > > > > ### Author Response · Authors · 2024-11-30
> > > > > > > > > > > > >
> > > > > > > > > > > > > We appreciate your thorough feedback regarding our experimental methodology. While we fully respect your assessment, as stated before, we must strongly disagree with the notion that our experimental design is insufficient for demonstrating our findings.
> > > > > > > > > > > > >
> > > > > > > > > > > > > *You only show one case which is with low-entropy, and one case which is with high-entropy, and claim low-entropy is more stealthy, the high-entropy is more effective, it is not enough for a solid research, and this simple conclusion, which is widely-known, can also be expected even before your experiments.*
> > > > > > > > > > > > >
> > > > > > > > > > > > > ANS: While we understand the desire for extensive experimentation, we believe there are several important considerations:
> > > > > > > > > > > > >
> > > > > > > > > > > > > 1. Our research demonstrates a **linear relationship between entropy gap and jailbreak effectiveness** - a finding that, while aligned with intuition, had not been empirically validated before. From a mathematical perspective, two points are sufficient to establish a linear relationship, and our selected high and low entropy cases serve this purpose effectively.
> > > > > > > > > > > > >
> > > > > > > > > > > > > 2. In the emerging field of AI safety research, where model behaviors and capabilities are rapidly evolving, establishing fundamental relationships can be more valuable than exhaustive testing of already-evident patterns. At least from our perspective, this focused approach to validating key relationships provides meaningful insights into jailbreak attacks.
> > > > > > > > > > > > >
> > > > > > > > > > > > > 3. The resource constraints faced by academic institutions are a real consideration that shouldn't invalidate meaningful research contributions. While industry labs may be capable of large-scale experiments, academic research often provides value through precise, well-designed studies illuminating specific phenomena.
> > > > > > > > > > > > >
> > > > > > > > > > > > > Our research makes a meaningful contribution to the field by providing theoretical and empirical evidence for this relationship, even if the general direction of the finding aligns with intuition. Not every valuable research contribution requires exhaustive experimentation, particularly when clear patterns emerge from well-designed, focused studies.

---

### Official Review · Reviewer_yeyL · 2024-11-04

**Soundness:** 1
**Presentation:** 1
**Contribution:** 2
**Rating:** 3
**Confidence:** 4

**Summary:**

This paper proposes a jailbreak detection as well as a jailbreak attack method targeting vision-language models. The detection method is based on a classifier trained on the intra-entropy gap of randomly partitioned parts of images. The attack pipeline is centered around a diffusion model with several pre-processing and post-checker modules. The evaluation shows the detection performance is not trivial, and the attack method is effective.

**Strengths:**

+ The paper discusses both detection and attack algorithms, which could be potentially more comprehensive than other papers.
+ The proposed algorithm is simple and straightforward to implement.

Studying the security issues of VLMs is a hot and relatively new area in the research community. This paper discussed the jailbreak problem from both attack and defense perspectives and tried to provide some formal analysis for their detection algorithm, which might contribute to the community if they could make more efforts in mathematical proof and algorithm design.

**Weaknesses:**

- The quality of this paper falls noticeably below the expected standard, and there are clear signs of last-minute efforts to meet the deadline, making the paper hard to read.
- The connection between the attack and defense algorithms is unclear, and I do not understand why the authors tried to integrate them into one paper.
- There might be some rigorous issues for the proposed method and mathematical proof.

I strongly encourage the authors to polish their papers before submitting them to top-tier conferences. Nowadays, we get tons of submissions from the community, and presenting something that is probably rejected has a negative influence on the paper-reviewing process.

One of the major limitations of the paper is its presentation. I am quite confused about what on earth the authors are trying to deliver. What is the current challenge and the research gap this paper wants to address? What is your methods' motivation (Your section 4 seems to jump from nowhere)? What is the point of section 3 with only around twenty lines? What is the structure of your evaluation, and do you want to emphasize attack or detection? The poor presentation of this paper makes it hard to follow, and I am quite confused about the major contribution of this paper.

While it might be good if the authors could introduce both attack and defense methods that achieve SOTA in one paper, I am afraid this is, in fact, very hard to achieve, and it turns out that the proposed methods have many problems.

First of all, the intra-entropy gap algorithm just does not make sense to me. It is based purely on random partition of the input into two non-overlapping regions. The authors did not provide formal discussions on why random partition can work and what the relationship is between the number of trials and the detection performance. As for theorem 1, the assumption on Markov chains is too strong, and proof of corollary 2 seems to be incorrect to me as Y2 = R1 + R2 does not imply anything about their entropy. As for the novelty, this random ablated then inference style detection and defense is common early in the pre-LLM ages, and I do not think the idea is new.

The diffusion-centric attack pipeline is more sound than the detection algorithm in some sense since it is a more engineering-style framework instead of an algorithm that requires mathematical proof. However, the design of the algorithm is rough and too simple, which makes it unsuitable to be published in top-tier AI conferences. The idea of turning text into images using the diffusion model for attack is also not novel, and several papers have already been published (e.g., [1]). What's worse, the performance of the proposed method is also limited.

Since both the detection and attack methods are not novel and not significant, and the formal analysis and evaluation results do not provide any intriguing insights, I tend to reject this paper.


[1] Xu, Wenzhuo, Kai Chen, Ziyi Gao, Zhipeng Wei, Jingjing Chen, and Yu-Gang Jiang. "Highly transferable diffusion-based unrestricted adversarial attack on pre-trained vision-language models." In ACM Multimedia 2024. 2024.

**Questions:**

What isWhat is the connection between your detection and attack methods?

How do you decide K (the number of trials) in your detection method?

---

> ### Author Response · Authors · 2024-11-21
>
> We appreciate the reviewer's insightful questions about the connection between our detection and attack methods and how to decide K (number of trials) in our method.
>
> 1. What is the current challenge and the research gap this paper wants to address? What is your methods' motivation?
>
>     **ANS:** We explicitly state our motivation in lines 58-68: "*significant research has focused on heuristic jailbreaking methods, yet the relationship between attack success rates and stealthiness remains unclear. We are the first to reveal an information-theoretical tradeoff between jailbreakability and stealthiness in VLMs.*" We will emphasiz and explain the research gap: current VLM jailbreak attack and detection methods both lack theoretical foundations and formal analysis. In addition, Figure 1 provides empirical evidence supporting this motivation by demonstrating the perplexity and entropy gaps between natural and jailbreak samples.
>
> 2.  What is the point of section 3 with only around twenty lines?
>
>     **ANS:** Section 3 presents the formal problem formulation of VLM jailbreaking. For brevity, extended preliminaries related to Section 3 are provided in Appendix C.4.
>
> 3.  What is the structure of your evaluation, and do you want to emphasize attack or detection?
>
>     **ANS:** As clearly indicated in our title and throughout the paper, our primary contribution is establishing the theoretical trade-off between jailbreakability and stealthiness. Both attack and detection methods serve as empirical validation of our theoretical framework, as evidenced by:
>      * Theorem 1 establishing the fundamental bounds.
>      * Corollary 2 connecting these bounds to practical detection metrics.
>      * Extensive experimental validation in Section 5. (e.g. While our detection framework successfully identifies other attack methods, it is less effective against our own attack. The modest performance improvements compared to baselines are primarily attributed to our emphasis on maintaining attack stealthiness.)
>
> 4. Does Intra-Entropy gap algorithm makes sense in both theory and practice?
>
>     **ANS:**
>     - First of all, we want to clarify that the random partition approach is actually principled - it's based on the theoretical insight that adversarial modifications tend to create **local inconsistencies** in image statistics.
>     - Secondly, the critique that Markov chain assumptions are overly restrictive does not invalidate their theoretical value. Indeed, the Markov property serves as a foundational principle in Reinforcement Learning (RL), enabling numerous practical algorithms. Moreover, theoretical frameworks inherently require abstraction to elucidate fundamental principles underlying real-world phenomena.
>     - Finally,  the equation Y2 = R1 + R2 alone does not directly imply entropic relationships. However, by applying the entropy function to both sides and utilizing relevant inequalities (detailed in Appendix B), the desired result follows. This validates the proof of Corollary 2.
>
> 5. Regarding to the writing of our paper:
>
>     **ANS:** Since the Reviewer AJaG commented "The paper is well-written, and the presentation is clean and clear," which aligns with our assessment, we may consider the manuscript's writing style is not a significant concern. In particular, after our responses to your comments, if there are questions that need to be further clarified, please let us know and we are happy to address any concerns.
>
> 6. What is the connection between your detection and attack methods?
>
>     **ANS:** Our attack and detection methods serve as empirical validation of the theoretical relationship. Specifically, the attack demonstrates efficacy in jailbreaking both open and closed source models while evading our detection framework. Conversely, the detection method successfully identifies non-stealthy jailbreak attacks.

---

> ### Author Response · Authors · 2024-11-21
>
> 7. How do you decide K (the number of trials) in your detection method?
>
>     **ANS:**
>     - Note that in line 198 we mention that these methods can be computationally intensive for large images. Therefore, we utilize rotation partitioning (Algorithm 3 in the Appendix D) for practical efficiency.
>     - While not central to our methodology, we can readily provide a formal analysis showing that K trials achieve probabilistic guarantees of detection with confidence $1-\delta,$ as demonstrated by the following analysis:
>
>         - **Detection Guarantee**
>         Let $I$ be an image with adversarial modifications affecting at least $\alpha$ fraction of the image area. For any $\delta > 0$, if we set $K = \left\lceil\frac{\log(1/\delta)}{\alpha}\right\rceil$ random trials in Algorithm 1, then the probability of failing to detect the modification is at most $\delta$.
>         **Proof**:
>         For each random partition $(R_1, R_2)$, the probability of the partition line intersecting the modified region is at least $\alpha$. Therefore, the probability of missing the modification in a single trial is at most $(1-\alpha)$. After $K$ independent trials, the probability of missing in all trials is at most $(1-\alpha)^K$. Setting $K = \left\lceil\frac{\log(1/\delta)}{\alpha}\right\rceil$ ensures:
>         \begin{align*}
>         (1-\alpha)^K &\leq \exp(-\alpha K) \\\\
>         &\leq \exp\left(-\alpha \cdot \frac{\log(1/\delta)}{\alpha}\right) \\\\
>         &= \exp(-\log(1/\delta)) \\\\
>         &= \delta
>         \end{align*}This implies that with $K$ trials, we detect the modification with probability at least $1-\delta$.
>
>             **Practical Detection Bound**
>             For a desired confidence level of $95\%$ ($\delta = 0.05$) and assuming the adversarial modification affects at least $10\%$ of the image ($\alpha = 0.1$), setting $K = 30$ trials is sufficient for reliable detection.
>             With $\alpha = 0.1$ and $\delta = 0.05$:
>             $$K = \left\lceil\frac{\log(1/0.05)}{0.1}\right\rceil = \left\lceil\frac{3}{0.1}\right\rceil = 30$$
>
> 8. Regarding novelty:
>
>     **ANS:** We are not claiming the idea of turning text into images using the diffusion model for attack is also novel; however, to our knowledge, we are the first to have a theoretical discussion on the tradeoff between jailbreakability and stealthiness. This theoretical result explains why our entropy-gap detection method (Algorithm 1) is effective: large entropy gaps indicate non-uniform information distribution characteristic of adversarial modifications. It also suggests that successful stealthy attacks should maintain similar entropy levels across image regions.

---

### Meta-Review · Area_Chair_9mJL · 2024-12-11

**Metareview:**

This paper reveals the trade-off between jailbreakability and stealthiness in VLMs, further motivating the development of detection and attack methods. Reviewers raised concerns about the paper's presentation, the clarity of its motivation, main technical contribution, and the comprehensibility of both the theoretical analysis and practical implications. Given these issues, the paper cannot be accepted to the conference in its current form. I recommend the authors continue improving the paper for future submissions.

**Additional Comments On Reviewer Discussion:**

None

---

### Decision · Program_Chairs · 2025-01-22

Reject